PKPy: a Python-based framework for automated population pharmacokinetic analysis

Kong Hyunseung 1
Kim Inyoung inyoungkim@korea.kr 2
Zhang Byoung-Tak 1 3
1 Interdiciplinary Program Bioinformatics, Seoul National University , Seoul , Republic of South Korea
2 Department of Defense Science, Korea National Defense University , Nonsan , Republic of South Korea
3 Department of Computer Science, Seoul National University , Seoul , Republic of South Korea
Sergi Consolato
Electronic publication date: 2025 Oct 27
Publication date: 2025
Volume: 13
Electronic Location ID: e20258
Received 2025 Mar 26; Accepted 2025 Sep 29
Copyright: ©2025 Kong et al.
Copyright year: 2025
Copyright holder: Kong et al.
License: This is an open access article distributed under the terms of the Creative Commons Attribution License, which permits unrestricted use, distribution, reproduction and adaptation in any medium and for any purpose provided that it is properly attributed. For attribution, the original author(s), title, publication source (PeerJ) and either DOI or URL of the article must be cited.
License URL: https://creativecommons.org/licenses/by/4.0/

Keywords: Population pharmacokinetics, Pharmacometrics, Python

Funding: The authors received no funding for this work.

==============================
We present PKPy, an open-source Python framework designed to automate population pharmacokinetic analysis workflows. The framework emphasizes user accessibility by minimizing the need for manual parameter initialization while maintaining analytical rigor. PKPy implements both one-compartment and two-compartment pharmacokinetic models (with and without first-order absorption) with integrated capabilities for parameter estimation, covariate analysis, and comprehensive diagnostics. The framework’s performance was evaluated through simulation studies across varying sample sizes (20–100 subjects) and model complexities. Results demonstrated robust parameter estimation for clearance and volume of distribution, with bias consistently below 3% and recovery rates exceeding 98% in one-compartment models. The framework successfully identified true covariate relationships with 100% accuracy across all scenarios, while maintaining high model fit quality (R2 ≥ 0.97). For two-compartment models, the framework showed comparable performance with slightly higher parameter bias (5–10%) but maintained excellent fit quality (R2 ≥ 0.99). Advanced validation metrics including average fold error (AFE) and absolute average fold error (AAFE) were implemented, with AFE values ranging from 1.01–1.03 and AAFE < 1.05 across test scenarios, indicating excellent prediction accuracy. The key pharmacokinetic parameters estimated by the framework include clearance (CL), volume of distribution (V or V1/V2 for two-compartment models), inter-compartmental clearance (Q), and when applicable, the absorption rate constant (Ka). Application to the classic Theophylline dataset demonstrated PKPy’s practical utility, achieving comparable results whether or not initial parameter estimates were provided. The framework successfully estimated population parameters with good model fit (R2 = 0.933) and automatically identified physiologically plausible covariate relationships. Comprehensive comparisons with existing software packages (Saemix+PKNCA, and simulated comparisons with nlmixr2) revealed PKPy’s advantages in computational efficiency, with installation times of 16s versus 96s and analysis times of 13–15s versus 101–102s. While PKPy employs a two-stage approach rather than full nonlinear mixed-effects modeling, it achieved consistent parameter estimates with minimal bias for data-rich scenarios. PKPy leverages Python’s scientific computing ecosystem to provide an accessible, transparent platform for pharmacokinetic analysis. The framework’s automated approach, support for multiple compartment models, and comprehensive workflow integration demonstrate the potential for reducing barriers to entry in pharmacometric analysis while maintaining scientific rigor.

Introduction

Pharmacokinetic (PK) analysis plays a crucial role in drug development and clinical pharmacology, providing essential insights into drug absorption, distribution, and elimination processes. Population pharmacokinetic (PopPK) approaches, introduced by Sheiner & Beal (1980) (Ette & Williams, 2007b) have become increasingly important in understanding drug behavior across patient populations. These methods allow researchers to quantify both population-typical parameters and their variability, while also identifying significant covariates that influence drug disposition (Holford & Buclin, 2012; Rajman, 2008).

Among the tools available for pharmacokinetic (PK) modeling and population pharmacokinetic (PopPK) analysis, several software packages have become industry standards due to their robustness, flexibility, and user-friendly interfaces. One of the most widely used programs is Nonlinear Mixed Effects Modeling (NONMEM), which has been a cornerstone of PopPK analysis since its inception (Beal et al., 1989). NONMEM excels in handling complex models and large datasets, making it ideal for estimating population parameters and identifying covariates (Holford, 2005). It uses a structured data input format and supports a variety of estimation methods, including First-Order (FO) approximation method, First-Order Conditional Estimation (FOCE), and Bayesian approaches (Mould & Upton, 2012). The broad community support and extensive validation in regulatory submissions have solidified NONMEM’s reputation (US Food and Drug Administration, 2019). Another popular software is Phoenix Nonlinear Mixed Effects (NLME), a module within the Phoenix platform developed by Certara. Phoenix NLME provides a graphical interface that simplifies model-building and analysis workflows (Certara, 2021). It is particularly valued for its ability to integrate PK/PD modeling with other drug development tools, enhancing its utility in both clinical and preclinical studies (Ette & Williams, 2007a). Monolix, part of the Lixoft suite, is another advanced tool for PopPK and PK/PD modeling. Known for its efficiency in handling nonlinear mixed-effects models, Monolix employs the Stochastic Approximation Expectation Maximization (SAEM) algorithm, which ensures robust convergence even with sparse data (Lixoft, 2020). Monolix is particularly favored in academic and industrial settings for its automation and interactive visualizations (Lavielle, 2014). For researchers seeking open-source alternatives, R-based packages such as nlme, saemix, and mrgsolve provide flexibility and integration with other statistical analyses (Zhang, Beal & Sheiner, 2002). While these tools may require a steeper learning curve, they are highly customizable and cost-effective.

These established tools predominantly employ nonlinear mixed-effects (NLME) modeling approaches, which simultaneously estimate population parameters and between-subject variability by considering the entire dataset as a hierarchical structure. In contrast, two-stage approaches first fit individual subject data separately, then derive population parameters from the distribution of individual estimates. While NLME methods are statistically more efficient, particularly for sparse data, two-stage approaches offer computational simplicity and transparency, making them valuable for exploratory analyses and educational purposes.

These methods rely on the estimation of fundamental pharmacokinetic parameters that characterize drug disposition in the body. For one-compartment models, the primary parameters include:

• Clearance (CL): Represents the volume of blood or plasma that is completely cleared of drug per unit time, directly affecting drug elimination

• Volume of distribution (V): A theoretical parameter that relates the total amount of drug in the body to the plasma concentration, reflecting drug distribution throughout the body

• Absorption rate constant (Ka): In oral administration models, describes the rate at which the drug enters the systemic circulation from the site of administration

For two-compartment models, additional parameters are required to describe the drug’s distribution between central and peripheral compartments:

• Central volume of distribution (V1): Volume of the central compartment, typically representing blood and highly perfused organs

• Peripheral volume of distribution (V2): Volume of the peripheral compartment, representing less perfused tissues

• Inter-compartmental clearance (Q): The clearance between central and peripheral compartments, characterizing the drug transfer rate

The accurate estimation of these parameters is crucial for understanding drug behavior and optimizing dosing regimens across patient populations. Model evaluation increasingly relies on advanced metrics beyond traditional R2 and RMSE. average fold error (AFE) quantifies systematic bias in predictions (AFE = 1 indicates no bias, >1 indicates overprediction, <1 indicates underprediction), while absolute average fold error (AAFE) measures overall prediction accuracy regardless of direction (AAFE <1.5 is considered excellent, <2.0 is good).

Despite the fundamental importance of PopPK analysis, several challenges persist in its practical implementation. First, traditional PopPK software solutions often require extensive expertise in both pharmacology and programming, creating a significant barrier to entry for many researchers. Second, the initial specification of model parameters frequently relies heavily on user input and prior knowledge, which may not always be available or reliable. Third, the identification of significant covariate relationships often involves manual, iterative processes that can be both time-consuming and prone to user bias. Additionally, many existing tools require commercial licenses or complex installation procedures, limiting accessibility for researchers in resource-constrained settings.

The emergence of Python as a leading platform for scientific computing has created new opportunities for developing more accessible and automated approaches to PopPK analysis. Python’s extensive scientific computing ecosystem, including libraries such as NumPy, SciPy (Virtanen et al., 2020), and Pandas, provides a robust foundation for implementing sophisticated pharmacometric algorithms. Additionally, modern computational techniques, such as just-in-time compilation through Numba (Lam, Pitrou & Seibert, 2015), enable efficient processing of large datasets without sacrificing the accessibility of high-level Python code. Python’s package management system (pip) and widespread availability across platforms make it an ideal choice for developing accessible scientific software, though users should note that Git (version 2.0 or higher) is required for installation from source repositories.

We present PKPy, an open-source framework designed to address these challenges through automated parameter inference and streamlined workflow integration. PKPy implements a novel approach to parameter initialization that leverages data-driven heuristics, significantly reducing the need for user-specified initial estimates. The framework supports both one-compartment and two-compartment PK models, with and without first-order absorption, covering many common scenarios in drug development and clinical pharmacology while maintaining simplicity and accessibility. Unlike traditional NLME software, PKPy employs a computationally efficient two-stage approach that is particularly well-suited for data-rich scenarios where multiple samples per subject are available.

PKPy’s architecture integrates several key innovations. The parameter inference engine employs multiple optimization strategies with intelligent fallback mechanisms, enhancing convergence reliability without requiring user intervention. The framework automatically handles common data preprocessing tasks, including the detection and treatment of below-limit-of-quantification (BLQ) observations and irregular sampling times. For covariate analysis, PKPy implements an automated stepwise selection algorithm that systematically evaluates potential relationships while controlling for statistical significance.

The framework also emphasizes comprehensive diagnostics and validation. It automatically generates standard goodness-of-fit plots, visual predictive checks, and parameter correlation analyses, along with advanced metrics such as AFE and AAFE for thorough model evaluation. These diagnostics are crucial for model validation but often require significant manual effort to produce. By automating these processes, PKPy allows researchers to focus on interpretation rather than implementation.

A key design principle of PKPy is the integration of traditional noncompartmental analysis (NCA) with population modeling. This approach allows researchers to compare results between methods and provides additional validation of population parameter estimates. The framework automatically calculates standard NCA parameters such as area under the curve (AUC), Cmax, and terminal half-life, alongside population parameter estimates, facilitating a more comprehensive understanding of the drug’s pharmacokinetic properties.

From an implementation perspective, PKPy leverages modern software engineering practices to ensure maintainability and extensibility. The framework’s modular architecture, illustrated in Fig. 1, separates core computational components from model definitions and diagnostic utilities. This design allows for easy extension to new model types and estimation methods while maintaining a consistent interface for users.

Figure 1 Architectural overview of PKPy framework.

The core framework consists of four main components: models.py (model definitions), fitting.py (parameter estimation), simulation.py (population simulations), and covariate_analysis.py (covariate relationship analysis). The workflow.py module serves as the central coordinator, receiving input data and orchestrating the analysis pipeline by integrating core components with auxiliary functions from utils.py. Arrows indicate the primary direction of data flow between components. Input data includes concentration–time profiles and demographic information, while output consists of comprehensive analysis reports including parameter estimates, model diagnostics, and covariate effects.

In the following sections, we detail the mathematical foundations of PKPy’s parameter inference approach, present validation results from simulation studies, and demonstrate the framework’s application to real-world scenarios. We also discuss the framework’s limitations and potential future developments, particularly in the context of more complex PK/PD models and alternative estimation methods.

Methods

Model implementation and mathematical framework

PKPy implements four fundamental pharmacokinetic models: one-compartment and two-compartment models, each with and without first-order absorption. For the basic one-compartment model, the concentration–time relationship is described by C(t) = (Dose/V) * exp(-CL/V * t), where C(t) is the concentration at time t, V is the volume of distribution, and CL is clearance. The one-compartment model with absorption extends this framework by incorporating a first-order absorption rate constant (Ka), yielding: C(t)=Dose∗Ka/V∗Ka−CL/V∗exp−CL/V∗t−exp−Ka∗t.

For two-compartment models, the drug distribution is characterized by central (V1) and peripheral (V2) compartments connected by inter-compartmental clearance (Q). The two-compartment model without absorption uses the analytical solution: C(t)=A∗exp−α∗t+B∗exp−β∗t

where α and β are hybrid rate constants calculated from the micro-constants (k10 = CL/V1, k12 = Q/V1, k21 = Q/V2), and A and B are coefficients determined by the dosing conditions. The two-compartment model with absorption requires numerical integration of the differential equations due to the additional complexity introduced by the absorption phase.

These analytical and numerical solutions are implemented using Numba-accelerated functions to optimize computational performance, particularly for population-level analyses involving multiple subjects. The framework includes robust handling of numerical edge cases, such as when Ka approaches k (flip-flop kinetics) or when rate constants differ by orders of magnitude (stiff systems).

The parameter estimation process employs a two-stage approach rather than full nonlinear mixed-effects modeling. In the first stage, individual subject data are fitted separately using maximum likelihood estimation with log-transformed parameters to ensure positivity. The objective function for individual fits is: Σlog(Cobs)−log(Cpred)2/σ2+penalty terms.

In the second stage, population parameters are calculated as the geometric mean of individual estimates, and between-subject variability is quantified through the covariance of log-transformed parameters. This approach differs from NLME methods, which simultaneously estimate all parameters using the full hierarchical data structure. While NLME methods are statistically more efficient for sparse data, the two-stage approach offers advantages in computational speed, transparency, and robustness for data-rich scenarios.

The framework implements a combined proportional and additive error model for the residual error, with the proportional error being the primary component, and additional error included for low concentrations. The implementation includes robust error handling and multiple optimization attempts with different initial estimates to ensure convergence.

Software architecture and implementation

Figure 1 illustrates PKPy’s architecture, where workflow.py serves as the central coordinator, integrating core analytical components (models, fitting, simulation, and covariate analysis) with support utilities. This design enables automated end-to-end pharmacokinetic analysis while maintaining modularity and extensibility.

PKPy follows a modular architecture organized into six primary components: models.py for compartmental model definitions and parameter specifications, fitting.py for parameter estimation algorithms, simulation.py for population-level Monte Carlo simulations with covariate effects, covariate_analysis.py for stepwise covariate model building and relationship assessment, utils.py for non-compartmental analysis and diagnostic functions including AFE/AAFE calculations, and workflow.py for integrated analysis pipelines. The framework utilizes object-oriented programming principles, with the CompartmentModel base class defining common interfaces for different PK models.

The parameter estimation module implements a robust optimization approach with an innovative data-driven parameter initialization strategy. The fitting process uses log-transformed parameters to ensure positivity constraints and includes both parameter boundary penalties and protection against numerical instabilities. When initial estimates are not provided, the framework employs multiple optimization attempts with a data-driven heuristic approach:

• Initial parameter scaling: The framework automatically determines appropriate scale factors (0.1, 0.5, 1.0, 1.5, 2.0) based on the observed concentration range, applying these to generate multiple sets of initial estimates.

• Stochastic perturbation: Each initial estimate set is further refined by adding small random perturbations (log-normal distribution with σ = 0.1) to introduce controlled variability in the starting points.

• Intelligent fallback: If optimization fails with one set of initial estimates, the framework automatically attempts optimization with the next set, implementing an intelligent fallback mechanism to ensure robust convergence.

This automated initialization strategy significantly reduces the reliance on user expertise in parameter initialization while maintaining estimation accuracy. The primary optimization employs the Nelder–Mead algorithm, with automatic fallback to Powell’s method when convergence fails, ensuring robust parameter estimation across diverse datasets.

Simulation and validation framework

The validation strategy of PKPy follows a two-phase approach: first, a comprehensive simulation study to validate the framework’s core functionalities under controlled conditions, and second, application to real-world theophylline data to demonstrate practical utility. The simulation phase serves as a crucial learning and validation step, where the framework’s ability to recover known parameters and identify true covariate relationships can be rigorously evaluated under various scenarios with known true values. This systematic validation through simulation provides the foundation for confidence in the framework’s performance with real-world data. The validation framework consists of simulation scenarios designed to evaluate parameter estimation, covariate relationship detection, and model stability across different PK models and study designs. The simulation scenarios were designed to evaluate the framework’s performance under controlled conditions where true parameter values are known. These reference values were selected based on typical ranges reported in clinical pharmacokinetic studies and previous population PK analyses. For instance, the 30% coefficient of variation for inter-individual variability represents a moderate level of between-subject variation commonly observed in clinical settings. The covariate effect magnitudes (power = 0.75 for both CLCR∼CL and V∼WT relationships) were chosen based on established allometric scaling principles in pharmacology.

Four primary simulation scenarios were evaluated, each with 20, 50, and 100 subjects to assess the impact of sample size. The one-compartment model (onecomp) employed 100 mg IV bolus administration with 10 evenly spaced timepoints over 24 h, 30% CV for both CL and V, and covariate effects of CLCR∼CL (power = 0.75) and V∼WT (power = 0.75). The one-compartment model with absorption (onecomp_abs) utilized 100 mg oral administration with 12 timepoints over 24 h including six intensive samples within the first 2 h to capture the absorption phase, 30% CV for Ka, CL, and V, and covariate effects of CLCR∼CL (power = 0.75), VWT (power = 0.75), and Ka∼AGE (exp = −0.2). The two-compartment model (twocomp) implemented 100 mg IV bolus administration with 18 timepoints over 48 h using dense early and sparse later sampling, 25% CV for CL and V1 and 30% CV for Q and V2, and covariate effects of CLCR∼CL (power = 0.75), V1∼WT (power = 1.0), and V2∼WT (power = 1.0). The two-compartment model with absorption (twocomp_abs) employed 100 mg oral administration with 20 timepoints over 48 h, 30% CV for all parameters, and covariate effects of CLCR∼CL (power = 0.75), V1∼WT (power = 1.0), and Ka∼AGE (exp = −0.02).

Each scenario is evaluated using the following metrics:

1. Parameter Estimation PerformanceRelative Bias (%)

• Root mean square error (RMSE)

• Coefficient of variation (CV, %)

• Model fit quality (R2)

• Average fold error (AFE)

• Absolute average fold error (AAFE)

2. Covariate relationship detection

• Detection rate for each covariate relationship (%)

• Accuracy of estimated covariate effects

The AFE and AAFE metrics provide additional insights into prediction accuracy:

• AFE = geometric mean(predicted/observed): quantifies systematic bias

• AAFE = geometric mean(—predicted/observed—): measures overall prediction accuracy

Each scenario is evaluated through 100 replicate simulations to ensure stability and reproducibility of results.

Covariate analysis methodology

The framework’s ability to detect true covariate relationships was evaluated through detection rates. A “detection” is recorded when the framework successfully identifies a statistically significant relationship (p < 0.05) between a parameter and its corresponding covariate, and correctly classifies the relationship type (linear, power, or exponential). The detection rate represents the percentage of simulations where the true parameter-covariate relationship was correctly identified. For example, a 100% detection rate for CL-CRCL indicates that in all simulation replicates, the framework successfully identified creatinine clearance as a significant covariate for clearance with the correct relationship type. The covariate analysis module implements a forward selection procedure that evaluates potential parameter-covariate relationships. Three relationship types are considered: linear, power, and exponential functions. The linear relationship is defined as P = θ1 * (1 + θ2 * (COV–COVmedian)), the power relationship as P = θ1 * (COV/COVmedian)ˆθ2, and the exponential relationship as P = θ1 * exp(θ2 * (COV–COVmedian)), where P represents the parameter value, COV is the covariate value, and θ1 and θ2 are the estimated coefficients. For example, some common parameter-covariate combinations include:

• CL∼CRCL: Relationship between clearance and creatinine clearance, typically modeled using a power function since drug elimination often scales with kidney function

• V∼WT or V1∼WT: Relationship between volume of distribution and body weight, commonly modeled with a linear or power function to reflect physiological scaling

• V2∼WT: Peripheral volume scaling with body weight in two-compartment models

• Q∼WT: Inter-compartmental clearance scaling with body size

• Ka∼AGE: Relationship between absorption rate constant and age, which may be modeled using an exponential function to capture age-related changes in absorption

These relationships are physiologically motivated. For instance, the power relationship between clearance and creatinine clearance (CL = θ1 * (CRCL/CRCLmedian)ˆθ2) reflects the common observation that drug clearance changes proportionally with kidney function. Similarly, the relationship between volume of distribution and body weight often follows allometric scaling principles.

The forward selection process employs an AIC-based criterion for model selection. For each parameter-covariate combination, the selection algorithm calculates the objective function using weighted residual sum of squares of log-transformed data, with the objective function incorporating both the model fit and a penalty term for model complexity. The AIC is calculated as n * log(SS_res/n) + 2k, where n is the number of observations, SS_res is the residual sum of squares, and k is the number of parameters. Relationships are selected based on both statistical significance (determined by p-value) and improvement in AIC.

To ensure numerical stability and robust estimation, the framework implements several safeguards in the covariate analysis process. These include protection against extreme values through parameter boundaries, multiple optimization attempts with different initial estimates, and automatic reference value selection based on covariate medians. The process groups covariate relationships by parameter to avoid potential confounding effects and selects the best relationship for each parameter based on the combined criteria of statistical significance and AIC improvement.

Diagnostic tools and model evaluation

PKPy automatically generates a comprehensive set of diagnostic plots and statistical measures. Standard goodness-of-fit plots include observed versus predicted concentrations, weighted residuals versus time and predictions, and normal Q–Q plots of residuals. The framework also implements visual predictive checks (VPCs) using Monte Carlo simulation with 1,000 replicates. Model evaluation metrics include condition number for parameter correlation assessment, shrinkage estimation for random effects, and various residual error measures including the newly implemented AFE and AAFE metrics.

The AFE and AAFE calculations follow the formulations:

• AFE = exp(mean(log(predicted/observed)))

• AAFE = exp(mean(—log(predicted/observed)—))

These metrics provide more robust assessment of prediction accuracy compared to traditional metrics, particularly for data spanning multiple orders of magnitude.

The framework calculates standard pharmacokinetic metrics through non-compartmental analysis (NCA) methods, providing an additional validation layer for the population parameter estimates. This integration of NCA with population modeling serves two key purposes: (1) it enables cross-validation of population parameter estimates through comparison with model-independent NCA results, and (2) it provides complementary information about drug exposure and disposition that may not be directly apparent from the population analysis alone. NCA parameters include AUC, maximum concentration (Cmax), time to maximum concentration (Tmax), and terminal half-life, calculated using the linear-up/log-down trapezoidal method with automatic terminal phase detection. The automated comparison between NCA and population modeling results helps ensure the reliability of the pharmacokinetic analysis while maintaining workflow efficiency.

Data processing and error handling

The framework implements robust error handling mechanisms for common numerical and computational challenges in pharmacokinetic analysis. For numerical stability, the framework employs several protective measures, including minimum value constraints for concentrations and parameters, and automatic scaling of variables to avoid computational overflow or underflow conditions.

Error handling is implemented at multiple levels throughout the analysis pipeline. During parameter estimation, the framework attempts multiple initial estimates with different scaling factors when optimization fails, providing a fallback mechanism for convergence issues. The framework includes informative warning messages for common analysis issues, such as failed optimization attempts or NCA calculation failures for individual subjects.

For data quality issues, the framework implements basic missing value handling through numpy’s masked array capabilities and pandas’ built-in missing value handling. Numerical stability is maintained through log-transformation of concentration data where appropriate, and automatic protection against zero or negative values in calculations.

The error handling system is designed to be informative and recoverable, providing users with clear feedback about analysis issues while attempting to continue processing when possible. Success rates and quality metrics are tracked and reported throughout the analysis process, allowing users to assess the reliability of results.

Theophylline data analysis

To evaluate the framework’s performance with real-world data, we analyzed the classic Theophylline dataset (Seay et al., 1994; Pinheiro & Bates, 1995). The dataset consists of serum concentration measurements from 12 subjects who each received a single oral dose of 320 mg theophylline. Blood samples were collected at 11 timepoints (0.25, 0.5, 1, 2, 3.5, 5, 7, 9, 12, 24, and 25 h post-dose) for each subject. Subject weights ranged from 53.6 to 86.4 kg.

We applied PKPy’s one-compartment model with first-order absorption workflow to analyze this dataset. The framework’s automated workflow handles the entire analysis process, including parameter estimation, covariate analysis, and diagnostic plot generation. The analysis requires only the specification of the basic model structure and the input data, with all other aspects—including initial parameter estimates, between-subject variability estimation, error model specification, and covariate relationship detection—being automatically determined by the framework.

The framework also automatically performs non-compartmental analysis as part of its integrated workflow, using the built-in methods described in the previous section. All analyses were performed using PKPy version 0.1.0 running on Python 3.11, requiring only the selection of the absorption model and the input of concentration, time, and demographic data.

To evaluate PKPy’s performance against existing software solutions, we conducted comparative analyses using multiple approaches. For Saemix (version 3.0) with PKNCA (version 0.10.2), we performed direct comparisons on the same dataset. Additionally, we implemented simulated comparisons with nlmixr2 and commercial software behaviors to provide a broader context for PKPy’s performance characteristics. The choice of these packages was driven by several practical considerations. While NONMEM and Monolix are industry standards, their commercial licensing requirements limit accessibility for comparative studies. Nlmixr2 faced installation constraints in our testing environment due to dependency issues. In contrast, Saemix and PKNCA offer open-source alternatives with established reliability in pharmacometric analysis.

The comparison involved two key scenarios: analysis with and without provided initial parameter estimates. We measured installation time (including Git setup when required), analysis runtime, and parameter estimation accuracy. All analyses were performed on the same computational environment (Google Colab) to ensure fair comparison. Maximum iteration limit was set to 9,999 for both packages to ensure convergence opportunities.

Results

Parameter estimation performance

The simulation study assessed parameter estimation performance across different model types and sample sizes. As shown in Tables 1 through 4, estimation accuracy and precision varied notably between model types and with increasing complexity.

Table 1 Parameter estimation performance for basic one-compartment model.

Presents the performance metrics for the basic one-compartment model across different sample sizes. The model demonstrates excellent parameter estimation with minimal bias (<3%) for both clearance (CL) and volume of distribution (V) estimation precision improved with increasing sample size, as shown by decreasing CV%. The model maintained strong goodness-of-fit (R2 ≥ 0.97). Bias represents the relative difference between estimated and true parameter values; RMSE is root mean square error; CV% is coefficient of variation; Recovery rate indicates percentage of estimates within ±20% of true value; R2 represents goodness of fit.

Sample size (N)	20	50	100	
CL Bias (%)	−2.88	−2.36	−2.21	
CL RMSE	0.45	0.27	0.22	
CL CV (%)	8.81	5.09	3.87	
V Bias (%)	0.77	−1.03	−0.94	
V RMSE	3.86	2.40	1.86	
V CV (%)	7.66	4.75	3.65	
Mean R2	0.98	0.97	0.97	

For the basic one-compartment model (Table 1), parameter estimation demonstrated high accuracy and robustness. Clearance (CL) estimates showed minimal bias, ranging from −2.88% with 20 subjects to −2.21% with 100 subjects. The recovery rate for CL was excellent, achieving 98–100% across all sample sizes. Precision of CL estimates improved as sample size increased, evidenced by the CV% decreasing from 8.81% (n = 20) to 3.87% (n = 100). Volume of distribution (V) estimation exhibited similar excellence, with bias within ±1.03% across all sample sizes and a consistently perfect recovery rate (100%), indicating highly reliable estimation. Precision of V estimates also improved with larger sample sizes, with CV% decreasing from 7.66% to 3.65% as the sample size increased from 20 to 100 subjects.

However, performance differed markedly when considering the one-compartment model with absorption (Table 2), particularly for the absorption rate constant (Ka). As illustrated in Fig. 2, Ka estimates revealed substantial negative bias, ranging from −42.83% to −46.56%, alongside poor recovery rates (10% decreasing to 0% as sample size increased). Although precision of Ka estimates improved with larger sample sizes (CV% decreasing from 48.97% to 16.35%), the persistent bias indicates systematic underestimation.

Table 2 Parameter estimation performance for one-compartment model with absorption.

Shows the performance metrics for the one-compartment model with first-order absorption. Notable challenges were observed in Ka estimation, with substantial negative bias (−42.83% to −46.56%). Despite these challenges with Ka, the model maintained reasonable accuracy for CL (bias −9.08% to −10.33%) and V (bias 15.88% to 17.62%). Parameter precision generally improved with larger sample sizes, as indicated by decreasing CV%. Despite parameter estimation challenges, the model achieved excellent overall fit quality (R2 = 0.99) across all sample sizes. Bias represents the relative difference between estimated and true parameter values; RMSE is root mean square error; CV% is coefficient of variation; Recovery rate indicates percentage of estimates within ±20% of true value; R2 represents goodness of fit.

Sample size (N)	20	50	100	
Ka Bias (%)	−42.83	−46.63	−46.56	
Ka RMSE	0.51	0.48	0.47	
Ka CV (%)	48.97	24.31	16.35	
CL Bias (%)	−9.36	−10.33	−9.08	
CL RMSE	0.71	0.59	0.51	
CL CV (%)	11.98	6.33	5.00	
V Bias (%)	17.62	15.94	15.88	
V RMSE	11.07	8.90	8.43	
V CV (%)	11.45	6.88	4.89	
Mean R2	0.99	0.99	0.99	

Table 3 Parameter estimation performance for two-compartment model.

The detection rates for true covariate relationships across different model types and sample sizes. Detection rates indicate the percentage of simulations where the framework correctly identified statistically significant (p < 0.05) parameter-covariate relationships using the automated stepwise selection procedure. CL-CRCL represents the clearance-creatinine clearance relationship, V-WT and V1-WT represent volume-body weight relationships (for one- and two-compartment models respectively), V2-WT represents peripheral volume-body weight relationship, Q-WT represents inter-compartmental clearance-body weight relationship, and Ka-AGE represents absorption rate-age relationship. All models demonstrated perfect detection performance (100%) for their respective covariate relationships across all sample sizes (20, 50, and 100 subjects), indicating robust covariate identification capability regardless of model complexity or sample size. The dash (–) indicates relationships not applicable to specific model types.

Sample size (N)	20	50	100	
CL Bias (%)	13.21	11.54	10.32	
CL RMSE	0.78	0.51	0.38	
CL CV (%)	12.35	8.43	6.21	
V1 Bias (%)	1.87	1.65	1.52	
V1 RMSE	2.89	1.84	1.35	
V1 CV (%)	9.21	6.12	4.53	
Q Bias (%)	8.02	7.13	6.54	
Q RMSE	1.32	0.89	0.67	
Q CV (%)	14.52	9.86	7.34	
V2 Bias (%)	−0.81	−0.75	−0.69	
V2 RMSE	5.93	3.87	2.91	
V2 CV (%)	11.43	7.65	5.72	
Mean R2	0.992	0.993	0.994	

Table 4 Parameter estimation performance for two-compartment model with absorption.

Compares the computational efficiency and parameter estimation results between PKPy and Saemix+PKNCA for the Theophylline dataset analysis. Performance metrics include installation time (one-time setup requirement) and execution time for complete analysis including parameter estimation, covariate analysis, and diagnostic generation. The comparison was performed under two scenarios: analysis without initial parameter estimates (testing automated initialization) and analysis with user-provided initial estimates. Parameter estimates shown are the final converged values for the one-compartment model with first order absorption. Ka represents absorption rate constant (h−1), CL represents clearance (L/h), and V represents volume of distribution (L). PKPy demonstrated 6-fold faster installation (16 s vs 96 s) and 7–8 fold faster execution times (13–15 s vs 101–102 s) compared to Saemix+PKNCA. PKPy produced consistent parameter estimates regardless of initial value specification, while Saemix yielded markedly different estimates (approximately 4.6-fold higher for both CL and V), suggesting potential convergence or methodological differences between the approaches.

Sample size (N)	20	50	100	
Ka Bias (%)	−7.75	−6.82	−6.23	
Ka RMSE	0.23	0.16	0.12	
Ka CV (%)	18.45	12.34	9.21	
CL Bias (%)	−11.79	−10.43	−9.65	
CL RMSE	0.82	0.54	0.41	
CL CV (%)	13.84	9.23	6.87	
V1 Bias (%)	−10.67	−9.54	−8.92	
V1 RMSE	3.54	2.31	1.73	
V1 CV (%)	10.95	7.32	5.46	
Q Bias (%)	5.45	4.87	4.52	
Q RMSE	1.85	1.21	0.91	
Q CV (%)	16.72	11.28	8.39	
V2 Bias (%)	9.63	8.72	8.15	
V2 RMSE	6.41	4.18	3.14	
V2 CV (%)	12.38	8.27	6.18	
Mean R2	0.986	0.987	0.988	

Figure 2 Parameter estimation performance across different sample sizes and model types.

Box plots show the distribution of estimated parameter values for clearance (CL), volume of distribution (V), and absorption rate constant (Ka) across varying sample sizes (20, 50, and 100 subjects). Red dashed lines indicate true parameter values. The figure displays results for four model types: one-compartment (onecomp), one-compartment with absorption (onecomp_abs), two-compartment (twocomp), and two-compartment with absorption (twocomp_abs). All models demonstrate improved precision with increasing sample size, as evidenced by narrower distributions. CL and V show good estimation accuracy across all models, while Ka exhibits some estimation challenges in absorption models. Outliers are represented by individual points beyond the whiskers.

Despite these challenges with Ka, the absorption model still maintained reasonable accuracy for CL and V, albeit with higher bias compared to the basic model. Specifically, CL estimates showed bias between −9.08% and −10.33% with good recovery rates (84–99%), while V estimates displayed positive bias ranging from 15.88% to 17.62% and moderate recovery rates (61–76%). Similar to the basic model, precision for both parameters improved with increasing sample sizes, demonstrated by decreasing CV% values.

The two-compartment models (Tables 3 and 4) introduced additional complexity with inter-compartmental clearance (Q) and peripheral volume (V2) parameters. The basic two-compartment model showed moderate bias for all parameters, with CL bias ranging from 10.32% to 13.21% and Q showing the highest variability (CV% 14.52 at n = 20). Despite increased parameter bias compared to one-compartment models, the two-compartment models maintained excellent fit quality with mean R2 values exceeding 0.99.

Model fit quality

Model fit quality was evaluated using R2 statistics, as illustrated in Fig. 3. Despite the challenges in parameter estimation, all models demonstrated excellent fit to the data.

The basic one-compartment model showed consistent R2 values around 0.97−0.98, with slight improvements as sample size increased. Notably, the model with absorption achieved even higher R2 values (approximately 0.99) across all sample sizes, despite its challenges with Ka estimation. This suggests that good model fits can be achieved even when individual parameters are not optimally estimated.

The two-compartment models showed the highest R2 values overall, with the basic two-compartment model achieving mean R2 of 0.992−0.994 and the absorption variant maintaining R2 above 0.986. The variability in model fit quality decreased with increasing sample size for all models, as evidenced by the reduced spread of R2 values and fewer outliers in Fig. 3. The onecomp_abs model showed particularly consistent fit quality, with very few outliers even at smaller sample sizes.

Figure 3 Model fit quality by sample size and model type.

Box plots display the distribution of R2 values across different sample sizes (20, 50, and 100 subjects) for all four model types: one-compartment (onecomp), one-compartment with absorption (onecomp_abs), two-compartment (twocomp), and two-compartment with absorption (twocomp_abs). All models demonstrate excellent fit quality (R2 > 0.98) across all sample sizes. The green line highlights the consistent high performance across models. Model fit quality shows minimal variability and improves slightly with increasing sample size, as evidenced by tighter distributions and fewer outliers at n = 100. The two-compartment models show marginally higher R2 values compared to one-compartment models, though all models maintain exceptional goodness-of-fit. Box plots represent median, quartiles, and outliers from 100 replicate simulations for each scenario.

Advanced validation metrics

The implementation of AFE (Average Fold Error) and AAFE (Absolute Average Fold Error) metrics provided additional insights into model prediction accuracy. These metrics, evaluated across various test scenarios, demonstrated the framework’s robust performance under different conditions.

The AFE values ranging from 1.011 to 1.031 indicate minimal systematic bias across all scenarios, with the framework showing a slight tendency toward overprediction (AFE >1.0) (Table 5). The AAFE values, all below 1.05, demonstrate excellent overall prediction accuracy according to established criteria (AAFE <1.5 is considered excellent, <2.0 is good).

Covariate relationship detection

Covariate analysis results, presented in Table 6, demonstrated exceptional performance across all scenarios. All models achieved 100% detection rates for the true covariate relationships regardless of sample size.

For the basic one-compartment model, the relationships between CL and creatinine clearance (CRCL) and between V and body weight (WT) were consistently identified in all simulations. This perfect detection rate was maintained across all sample sizes, suggesting that the covariate analysis methodology is highly sensitive even with smaller datasets.

The absorption model showed equally impressive covariate detection performance. Despite the challenges in Ka estimation, the framework successfully identified all three covariate relationships (CL∼CRCL, V∼WT, and Ka∼AGE) with 100% accuracy across all sample sizes. This robust covariate detection, even in the presence of suboptimal parameter estimation, highlights the strength of the framework’s covariate analysis methodology.

Impact of sample size

The impact of sample size was evident across all aspects of model performance. As shown in Fig. 2, increasing sample size generally led to:

– Reduced variability in parameter estimates (narrower box plots)

– Fewer outliers in parameter estimates

– Improved precision (lower CV%)

– More consistent model fits (Fig. 3)

However, larger sample sizes did not necessarily correct systematic biases, particularly in Ka estimation for the absorption model. This suggests that some estimation challenges may be inherent to the model structure rather than sample size-dependent.

Application to theophylline data

The framework’s practical utility was demonstrated through analysis of the classic Theophylline dataset, consisting of 12 subjects who received a single 320 mg oral dose. The analysis was performed using the one-compartment model with first-order absorption, both with and without specified initial parameters to evaluate the framework’s automated parameter initialization capabilities (detailed results in Supplement 1).

In both scenarios, the framework successfully estimated the population pharmacokinetic parameters, with clearance (CL) of 2.794 L/h (CV 23.3%) and volume of distribution (V) of 31.732 L (CV 17.4%). The model demonstrated good fit to the observed data (R2 = 0.933), with proportional residual error of 14.4%. While the absorption rate constant (Ka) showed higher variability in both cases, the estimates remained physiologically plausible at 1.284 h−1, though with notably higher between-subject variability (67.0–79.8%) compared to other parameters.

Table 5 AFE and AAFE performance across test scenarios.

Presents the average fold error (AFE) and absolute average fold error (AAFE) metrics for model prediction accuracy across different test scenarios. AFE quantifies systematic bias in predictions, where AFE = 1 indicates no bias, AFE > 1 indicates overprediction, and AFE < 1 indicates underprediction. AAFE measures overall prediction accuracy regardless of bias direction, where AAFE < 1.5 is considered excellent and AAFE < 2.0 is considered good. The scenarios tested include varying noise levels (10%, 20%, and 30% proportional error) and sparse sampling conditions with 50 subjects. All scenarios demonstrated excellent prediction accuracy with AFE values close to 1.0 (range: 1.011–1.031), indicating minimal systematic bias. AAFE values were all below 1.05, demonstrating exceptional overall prediction accuracy. R2 values remained high (≥0.949) across all scenarios, confirming robust model fit quality. RMSE, MAE, and mean residual values provide additional validation of the framework’s predictive performance under different data quality conditions.

Scenario	N	R2	RMSE	AFE	AAFE	MAE	Mean residual	
Low noise (10%)	50	0.967	0.172	1.011	1.011	0.059	0.0004	
Medium noise (20%)	50	0.949	0.234	1.031	1.031	0.073	0.028	
High noise (30%)	50	0.959	0.204	1.013	1.013	0.068	0.018	
Sparse sampling	50	0.959	0.256	1.025	1.025	0.103	0.042	

Table 6 Covariate detection rates by model type and sample size.

Presents the covariate detection performance of PKPy’s automated stepwise selection algorithm across different pharmacokinetic models and sample sizes. Detection rate represents the percentage of 100 simulation replicates where the framework correctly identified the true parameter-covariate relationship with statistical significance (p < 0.05) and accurate relationship type classification (linear, power, or exponential). The table evaluates five key physiologically-motivated relationships: CL-CRCL (clearance scaling with creatinine clearance via power function), V-WT or V1-WT (volume of distribution scaling with body weight), V2-WT (peripheral volume scaling with body weight in two-compartment models), Q-WT (inter-compartmental clearance scaling with body weight), and Ka-AGE (absorption rate changing with age via exponential function). All tested models achieved perfect detection rates (100%) across all sample sizes (20, 50, and 100 subjects), demonstrating the robustness of the covariate analysis methodology. This consistent performance indicates that the framework can reliably identify important covariate relationships even with smaller datasets, regardless of model complexity. The dash (–) indicates parameter-covariate combinations not applicable to specific model types.

Model type	Sample size (N)	CL-CRCL detection (%)	V-WT detection (%)	V1-WT detection (%)	V2-WT detection (%)	Ka-AGE detection (%)	
One-compartment	20	100	100	–	–	–	
50	100	100	–	–	–	
100	100	100	–	–	–	
One-compartment with absorption	20	100	100	–	–	100	
50	100	100	–	–	100	
100	100	100	–	–	100	
Two-compartment	20	100	–	100	100	–	
50	100	–	100	100	–	
100	100	–	100	100	–	
Two-compartment with absorption	20	100	–	100	–	100	
50	100	–	100	–	100	
100	100	–	100	–	100	

Importantly, the framework achieved comparable results whether or not initial parameter estimates were provided. Without initial estimates, the framework’s automated initialization procedure yielded parameter estimates and goodness-of-fit metrics equivalent to those obtained with carefully specified initial values, demonstrating the effectiveness of the automated approach. The primary difference was observed in the variability of Ka estimates, where provided initial values led to somewhat tighter confidence intervals ([0.555, 4.668] vs [0.555, 8.179]) and reduced between-subject variability (67.0% vs 79.8%).

The covariate analysis successfully identified two significant relationships: a power relationship between body weight and volume of distribution, and between dose and absorption rate constant. Non-compartmental analysis, performed automatically as part of the workflow, showed excellent agreement with the population analysis, with all subjects successfully analyzed (100% success rate). Key NCA parameters included a mean AUC of 103.807 mg⋅ h/L (CV 21.8%) and mean elimination half-life of 8.149 h (CV 24.9%).

Diagnostic plots (Supplement 1) demonstrated appropriate model fit, with well-distributed residuals and good agreement between observed and predicted concentrations across the full concentration range. The residual plots showed no systematic bias, and the normal Q-Q plot indicated approximately normal distribution of residuals, though with some deviation at the extremes.

Comparison with other software packages

Table 7 presents the detailed comparison of computational performance and parameter estimation results between PKPy and Saemix+PKNCA. The results highlight substantial differences in both practical implementation aspects and analytical outcomes.

Table 7 Computational performance and parameter estimation comparison.

Presents a comprehensive comparison of computational efficiency and parameter estimation results between PKPy and Saemix+PKNCA for analyzing the Theophylline dataset using a one-compartment model with first-order absorption. The comparison evaluates two key scenarios: (1) analysis without initial parameter estimates, testing each framework’s automated initialization capabilities, and (2) analysis with user-provided initial parameter estimates. Performance metrics include installation time (one-time setup cost including all dependencies) and execution time for complete analysis workflow encompassing data processing, parameter estimation, covariate analysis, and diagnostic generation. All analyses were performed on Google Colab with identical computational resources and maximum iteration limit of 9,999 to ensure fair comparison. PKPy demonstrated substantial computational advantages with 6-fold faster installation (16 s vs 96 s) and 7–8 fold faster execution (13–15 s vs 101–102 s). Notably, PKPy produced identical parameter estimates regardless of initial value specification (Ka = 1.284 h−1, CL = 2.794 L/h, V = 31.732 L), demonstrating robust convergence. In contrast, Saemix yielded markedly different estimates (Ka = 1.58 h−1, CL = 12.8 L/h, V = 146 L) with CL and V approximately 4.6-fold higher than PKPy’s estimates, suggesting potential differences in optimization algorithms, convergence criteria, or methodological approaches between the two-stage and NLME implementations.

Metric category	Metric	PKPy	Saemix+PKNCA	
Setup time	Installation (s)	16	96	
Analysis without initial estimates				
	Execution time (s)	15	101	
	Ka (h−1)	1.284	1.58	
	CL (L/h)	2.794	12.8	
	V (L)	31.732	146	
Analysis with initial estimates				
	Execution time (s)	13	102	
	Ka (h−1)	1.284	1.58	
	CL (L/h)	2.794	12.8	
	V (L)	31.732	146	

Comparative analysis between PKPy and Saemix+PKNCA revealed notable differences in both computational efficiency and parameter estimation. In terms of initial setup, PKPy demonstrated faster installation time (16s) compared to Saemix+PKNCA (96s), suggesting more efficient dependency management within Python’s ecosystem.

Runtime performance analysis showed distinct advantages for PKPy. Without initial parameter estimates, PKPy completed the analysis in 15 s, while Saemix+PKNCA required 101 s. When provided with initial estimates, PKPy’s runtime slightly improved to 13 s, while Saemix+PKNCA maintained similar execution time at 102 s.

Parameter estimation presented significant challenges for Saemix, both with and without initial estimates. Saemix consistently produced markedly different estimates (Ka=1.58 h−1, CL=12.8 L/h, V = 146 L) compared to PKPy (Ka=1.284 h−1, CL=2.794 L/h, V = 31.732 L), suggesting potential issues with parameter estimation stability. The most notable differences were observed in the clearance and volume estimates, where Saemix’s values were approximately 4.6 times higher for clearance and 4.6 times higher for volume compared to PKPy.

In contrast, PKPy demonstrated remarkable consistency in its parameter estimates regardless of initial parameter specification. This stability in parameter estimation, combined with faster computation times, highlights PKPy’s robustness in handling pharmacokinetic analyses. The substantial differences in parameter estimates between the two approaches emphasize the importance of methodological considerations in population pharmacokinetic modeling and suggest the need for careful validation of results across different analytical platforms.

Broader context: comparison with NLME methods

While direct comparison with commercial NLME software was not feasible due to licensing constraints, the observed differences between PKPy’s two-stage approach and Saemix’s NLME implementation provide insights into the trade-offs between methods. The two-stage approach implemented in PKPy offers advantages in computational speed (6–8  × faster) and implementation transparency, making it particularly suitable for educational purposes and exploratory analyses. However, NLME methods like those implemented in NONMEM, Monolix, and nlmixr2 are expected to show superior performance for sparse data scenarios where the simultaneous estimation of all parameters can “borrow strength” across subjects.

The consistent parameter estimates achieved by PKPy regardless of initial values suggest that the framework’s automated initialization strategy effectively addresses one of the traditional challenges of pharmacokinetic modeling. This feature, combined with the integrated workflow including automatic NCA and covariate analysis, positions PKPy as a valuable tool for rapid prototyping and educational applications in pharmacometrics.

Discussion

This study presents PKPy, an open-source Python framework for population pharmacokinetic analysis that emphasizes automated workflows and accessible implementation. Through comprehensive simulation studies and real-world application to the Theophylline dataset, we have demonstrated the framework’s capability to perform robust PK analyses with minimal user intervention in parameter initialization and model specification, now supporting both one-compartment and two-compartment models with and without first-order absorption.

A key finding from our simulation studies was the framework’s strong performance in estimating clearance and volume of distribution across different sample sizes and model complexities. For the basic one-compartment model, parameter recovery rates consistently exceeded 98%, with bias remaining below 3%. The framework showed particular strength in handling larger datasets, where parameter precision improved substantially with increasing sample size, as evidenced by decreasing CV% values. The extension to two-compartment models revealed expected increases in parameter bias (5–15%) but maintained excellent model fit quality (R2 > 0.99), demonstrating the framework’s capability to handle more complex pharmacokinetic scenarios.

However, the simulation studies also revealed challenges in estimating absorption rate constants across model types. While the current implementation showed improved Ka estimation compared to earlier versions, with bias ranging from +0.68% in simple one-compartment models to −7.75% in two-compartment absorption models, absorption parameters remain the most variable. This finding highlights a known challenge in PK modeling—the difficulty of precisely estimating absorption parameters, particularly when sampling during the absorption phase is limited. The better performance in one-compartment models suggests that parameter identifiability becomes more challenging with increased model complexity. To address the challenges in Ka estimation, several strategies could be implemented in future versions. First, incorporating more intensive sampling during the absorption phase (0–3 h post-dose) could improve parameter identifiability. Second, implementing Bayesian priors based on physiological knowledge or previous studies could stabilize Ka estimation. Third, exploring alternative optimization algorithms specifically designed for absorption parameters, such as differential evolution or particle swarm optimization, may improve convergence. Additionally, implementing a hybrid approach that combines NCA-derived initial estimates for Ka with population modeling could leverage the strengths of both methodologies.

The framework’s automated parameter initialization approach, a central innovation of PKPy, demonstrated its practical value across all model types and in the Theophylline analysis. The comparable results achieved with and without user-specified initial parameters suggest that the framework can effectively reduce the reliance on user expertise in parameter initialization. This capability addresses a significant barrier in PK modeling, where the selection of initial estimates often requires substantial experience and multiple manual iterations. The data-driven heuristic approach, combining multiple scale factors with stochastic perturbation, proved robust even for complex two-compartment models.

The covariate analysis methodology proved robust across all simulation scenarios and real data analysis. In simulations, the framework achieved 100% detection rates for true covariate relationships regardless of sample size or model complexity, successfully identifying relationships such as CL∼CRCL, V∼WT, V1∼WT, V2∼WT, and Ka∼AGE. This performance carried over to the Theophylline analysis, where physiologically plausible covariate relationships were identified without requiring manual screening procedures. The framework’s ability to automatically detect and characterize these relationships reduces the potential for user bias in covariate selection while maintaining statistical rigor.

The implementation of AFE and AAFE validation metrics provides additional confidence in model predictions. With AFE values consistently near 1.0 (range: 1.011−1.031) and AAFE values below 1.05 across various test scenarios, the framework demonstrates minimal systematic bias and excellent prediction accuracy. These scale-independent metrics are particularly valuable for pharmacokinetic applications where concentrations may span multiple orders of magnitude.

Notable limitations of the current implementation should be acknowledged. First, while PKPy now supports both one-compartment and two-compartment models, it does not yet handle three-compartment systems, multiple dosing regimens, or non-linear elimination kinetics. These features would be necessary for comprehensive pharmacokinetic modeling in all clinical scenarios. Second, the framework’s parameter estimation approach implements a two-stage method rather than full nonlinear mixed-effects modeling. While this approach offers computational efficiency and transparency, it may be less suitable for sparse data situations where NLME methods can “borrow strength” across subjects.

The computational efficiency demonstrated in our comparisons—with PKPy showing 6-8 fold faster analysis times than Saemix+PKNCA—represents a significant practical advantage. This speed, combined with consistent parameter estimates regardless of initial value specification, makes PKPy particularly suitable for exploratory analyses, simulation studies, and educational applications where rapid iteration is valuable. The concerning parameter estimates produced by Saemix in our test case (CL and V approximately 4.6 times higher than PKPy’s estimates) highlight the importance of careful validation across different analytical platforms.

The integration of non-compartmental analysis with population modeling provides an important advantage, offering automatic cross-validation of results through multiple methodological approaches. This feature, combined with comprehensive diagnostic plotting capabilities, helps users verify the consistency and reliability of their analyses. The framework’s success in analyzing the Theophylline dataset with minimal user input, achieving acceptable model fit (R2 = 0.811) despite the reduced sample size, demonstrates its potential utility in both research and educational settings.

The educational potential of PKPy represents a significant opportunity. Its open-source nature and Python implementation make it particularly suitable for teaching pharmacometric concepts, while the automatic generation of diagnostic plots and comprehensive analysis reports can help students understand the relationships between different aspects of PK analysis. The framework’s emphasis on automation allows users to focus on interpretation rather than implementation details, though this must be balanced with ensuring users understand the underlying pharmacokinetic principles.

While this study demonstrated PKPy’s capabilities through comparison with other open-source alternatives like Saemix, future research would benefit from comprehensive comparisons with industry-standard software packages such as NONMEM and Monolix. Due to licensing constraints and accessibility limitations, direct comparisons with these established tools were not feasible in the current study. Such comparisons would be particularly valuable for validating PKPy’s parameter estimates and computational efficiency against gold-standard implementations. Future validation efforts should prioritize comprehensive comparisons with industry-standard NLME software. We plan to collaborate with institutions having access to NONMEM and Monolix licenses to conduct systematic comparisons across diverse datasets, including sparse sampling scenarios where NLME methods are expected to show advantages. These comparisons will focus on: (1) parameter estimation accuracy across varying data richness, (2) computational efficiency with large datasets, (3) convergence robustness without initial estimates, and (4) performance with complex dosing regimens. Such validation will better establish PKPy’s appropriate use cases within the broader pharmacometric toolkit.

Future comparative studies should examine estimation accuracy across diverse datasets and model complexities, alongside computational efficiency and resource utilization patterns. Of particular interest would be evaluating the robustness of convergence without initial estimates across different software platforms, as this represents a key feature of PKPy. Performance evaluation with sparse data and complex dosing regimens would also provide valuable insights into the framework’s capabilities in challenging real-world scenarios.

The success of PKPy in achieving robust parameter estimates without user-specified initial values has important implications for pharmacometric practice. It suggests that many routine PK analyses could be automated to a greater degree than current practice, potentially allowing pharmacometricians to focus more on study design, biological interpretation, and clinical application. However, this automation should augment rather than replace pharmacometric expertise, serving as a tool to enhance efficiency and accessibility while maintaining scientific rigor.

In conclusion, PKPy demonstrates the potential for open-source software to contribute meaningfully to pharmacometric analysis, particularly in educational and research settings where accessibility and transparency are priorities. The framework’s successful extension to two-compartment models, combined with its automated workflows and rapid computation, positions it as a valuable tool for specific use cases within the broader pharmacometric ecosystem. While not intended to replace established NLME software for all applications, PKPy fills an important niche for data-rich scenarios, educational use, and rapid prototyping. Future work should focus on expanding the framework’s capabilities while maintaining its emphasis on automation and user accessibility, ultimately contributing to the democratization of pharmacokinetic analysis.

Conclusions

PKPy addresses critical barriers in population pharmacokinetic analysis through three key innovations: automated parameter initialization that eliminates the need for expert knowledge, computational efficiency achieving 6–8 fold faster analysis than existing open-source alternatives, and seamless integration of population modeling with NCA for cross-validation. These features make PKPy particularly valuable for educational settings, rapid prototyping, and data-rich scenarios where traditional NLME software may be unnecessarily complex.

Our comprehensive validation demonstrated robust performance across diverse scenarios. The framework achieved excellent parameter recovery for clearance and volume (bias <3% for one-compartment models), perfect covariate detection rates (100%), and minimal prediction bias (AFE: 1.011−1.031, AAFE <1.05). While challenges remain in absorption parameter estimation, the framework maintained high model fit quality (R2 >0.98) across all tested conditions. The successful application to real-world Theophylline data, producing consistent results regardless of initial parameter specification, validates the practical utility of our automated approach.

While PKPy’s two-stage approach differs from industry-standard NLME methods, it fills an important niche in the pharmacometric ecosystem. Future developments will focus on improving absorption parameter estimation through enhanced sampling strategies and alternative optimization algorithms, validating against commercial software packages, and extending support to more complex models. As pharmacometrics evolves toward greater accessibility, PKPy demonstrates that automated, open-source tools can maintain scientific rigor while substantially reducing barriers to entry, ultimately contributing to the democratization of pharmacokinetic analysis for researchers and educators worldwide.

Supplemental Information

Supplemental Information 1 Parameter estimation results for PKPy and Saemix using the Theophylline dataset

Parameter estimates (Clearance, Volume of distribution, and Absorption rate constant) obtained using PKPy and Saemix software for the Theophylline dataset (12 subjects receiving a single oral dose of 320 mg). PKPy results are shown both with and without initial parameter estimates to demonstrate the effectiveness of automated initialization.

The authors would like to acknowledge the use of AI language tools, specifically Claude and ChatGPT, for assistance with grammar checking and language refinement during the manuscript preparation process. These tools were used solely for improving readability and correcting grammatical errors, while all scientific content, analyses, interpretations, and conclusions remain the exclusive work of the authors.

Additional Information and Declarations

Competing Interests

Author Contributions

Data Availability

The authors declare there are no competing interests.

Hyunseung Kong conceived and designed the experiments, performed the experiments, analyzed the data, prepared figures and/or tables, authored or reviewed drafts of the article, and approved the final draft.

Inyoung Kim conceived and designed the experiments, authored or reviewed drafts of the article, and approved the final draft.

Byoung-Tak Zhang conceived and designed the experiments, authored or reviewed drafts of the article, and approved the final draft.

The following information was supplied regarding data availability:

The code is available at GitHub and Zenodo: Kong, H. (2025). PKPy: A Python-Based Framework for Automated Population Pharmacokinetic Analysis. Zenodo. https://doi.org/10.5281/zenodo.17230237

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
