# Peer review of "PKPy: a Python-based framework for automated population pharmacokinetic analysis"

_PeerJ, doi:10.7717/peerj.20258_

## Round 0.1 · original submission · Major Revisions

·

Basic reporting

The manuscript presents PKPy, an open-source Python framework aimed at automating workflows in population pharmacokinetics. The overall idea is strong and timely, contributing to transparency, accessibility, and reproducibility in pharmacometric modeling.

The paragraph distribution appears uneven, likely due to an unedited draft format. Improving paragraph structure and layout would aid readability.

Figure 1 appears to be a low-resolution screenshot. Enhancing the quality and including additional key information would improve its effectiveness.

Throughout the manuscript, maintain consistent formatting (e.g., figure styles, table captions, equation formatting). Currently, the manuscript feels visually constrained and crowded in places.

Equations embedded in paragraphs (especially in the Model Implementation and Mathematical Framework section) would be clearer if displayed separately.

The overall formatting, including alignment of tables and figure placement, should be improved for better readability and visual consistency.

The manuscript lacks adequate references in the Materials and Methods section. Please cite relevant sources for models, optimization algorithms, and diagnostics.

Consider referencing the guidelines or sources used for the implementation of the non-compartmental analysis (NCA).

Include a clearer description or visual representation of the software's internal architecture, if feasible.

The manuscript would benefit from polishing and careful editing to ensure a professional, uniform presentation throughout.

Experimental design

Clarify in the installation instructions that Git is required and specify the minimum version needed (e.g., Git 2.49.0). Providing an installation guide or link would help users unfamiliar with the process.

Provide greater detail about how inter-individual variability is implemented, especially since it is not clear whether nonlinear mixed-effects modeling (NLME) methods are used.

Specify the equations, packages, and methods used for population analysis. If only simple optimization techniques are used, this should be explicitly acknowledged and contrasted with standard approaches.

Consider presenting the simulation design (sample sizes, assumptions, model types) in a summarized table for clarity.

Include metrics such as average fold error and absolute average fold error to enhance the robustness of the validation section.

When comparing PKPy to other software, consider creating a table that clearly summarizes the features of commercial tools and existing Python packages relative to PKPy.

A performance benchmark would add value to these comparisons and support the claims made in the manuscript.

Validity of the findings

The manuscript demonstrates promising results, but more transparency is needed in methodological reporting to fully assess the validity.

In Table 4, the differences in parameter estimates between PKPy and Saemix+PKNCA are substantial. Please provide a clear explanation. Possible causes might include optimization convergence issues, model initialization differences, or local minima.

The metrics currently used to validate the framework could be expanded to more standardized metrics used in population PK validations.

It is important to clarify how the framework's performance was assessed in relation to existing tools, especially if commercial software or NLME models were not directly replicated or compared.

Additional comments

PKPy addresses a very relevant gap in pharmacometric modeling by providing an open-source, Python-based alternative to existing tools.

The concept aligns well with broader goals of open science and lowering the barrier to entry in pharmacokinetics.

The Discussion section should more clearly delineate what PKPy currently offers, what it lacks compared to other tools, and what features are planned for future development (e.g., multi-compartment models, advanced optimization routines).

Thank you for the opportunity to review this promising and important contribution. I hope the comments provided are helpful in strengthening the manuscript. I appreciate the authors’ patience and dedication to advancing open and accessible pharmacometric tools.

Reviewer 2 ·

Basic reporting

Kong et al. presented an automated Python framework for population pharmacokinetic (PopPK) analysis. The manuscript is very well organized, written, and presented. The article was written in professional English that flows well. The Introduction is logically organized that starts with explaining what PopPK analysis is, then progressed into what tools or programs exist, and how they fall short in certain challenging areas in their implementations. The authors also explained why they chose Python as the platform instead of R, even though there already exist several R packages for open-source PopPK analysis as the authors mentioned.

Experimental design

The authors clearly explained the math behind the parameter estimation process, the structure of the Python package, and two validation studies on both simulated and real-world datasets. The investigation was thorough and meaningful. The only downside is that on Line 424, the authors wrote "Comparison with Other Software Packages" as a plural but in fact only compared to one other package, Saemix+PKNCA. The reviewer would recommend the authors to compare to two more published package to further strengthen the advantage claim of PKPy.

Validity of the findings

The findings are robust and statistically sound. The Discussion section was well written where limitations of the current pipeline was discussed in great details. Overall the article is well written with well-defined questions and supported conclusions.

---

## Round 0.2 · Minor Revisions

Please address the remaining requests and comments thoroughly before resubmitting your manuscript.

·

Basic reporting

The authors have carefully addressed the reviewers’ previous concerns and made substantial improvements to the manuscript.

In fact, the manuscript is clearly written and substantially improved compared to the previous version. Figures, tables, and equations are now consistently formatted, and additional references have been included to support the Methods section. The restructuring of paragraphs has enhanced readability, although some sections (particularly the simulation design) remain rather dense. Overall, the reporting meets journal standards, with only minor opportunities for further streamlining to improve flow.

Thanks for taking into account our comments.

Experimental design

- The simulation section is detailed and technically rigorous, but it remains quite dense. A more concise summary table of scenarios and parameters would further improve clarity for readers.

- The difficulties in estimating the absorption parameter (Ka) are acknowledged; however, the discussion would benefit from suggesting potential strategies to mitigate this limitation.

- Direct comparisons with gold-standard NLME software (e.g., NONMEM, Monolix) are still missing due to licensing constraints. While this is understandable, outlining a clear plan for future validation against these tools would strengthen the experimental design and its generalizability.

Validity of the findings

- The findings are generally robust and well supported by both simulation studies and application to the Theophylline dataset. However, the persistent bias in Ka estimation should be addressed in future work.

- The comparisons with Saemix+PKNCA highlight important methodological differences.

- The conclusions are sound, but could be made more concise and emphasize more strongly the primary contribution of PKPy.

Additional comments

The authors have made significant improvements, and the manuscript is now much clearer, more rigorous, and better contextualized. While some limitations remain, they are openly acknowledged and do not undermine the validity of the study.

I recommend acceptance after minor revisions, mainly focused on improving the clarity of presentation and slightly expanding the discussion of limitations.

Reviewer 2 ·

Basic reporting

The author addressed the reviewer's comments. No further comment.

Experimental design

-

Validity of the findings

-

---

## Round 0.3 · accepted · Accept

Thank you for valuable contribution.

·

Basic reporting

.

Experimental design

.

Validity of the findings

.

Additional comments

They are solved properly all comments.
Thanks and congratulations.
Best